# FedCDA: Federated Learning with Cross-round Divergence-aware Aggregation

**Haozhao Wang[1], Haoran Xu[2], Yichen Li[3], Yuan Xu[1], Ruixuan Li[3,*], Tianwei Zhang[4]**
[1]S-Lab, Nanyang Technological University        [2]Zhejiang University
[3]Department of Computer Science, Huazhong University of Science and Technology
[4]Nanyang Technological University
{hz_wang, rxli}@hust.edu.cn, tianwei.zhang@ntu.edu.sg

## Abstract

In Federated Learning (FL), model aggregation is pivotal. It involves a global server iteratively aggregating client local trained models in successive rounds without accessing private data. Traditional methods typically aggregate the local models from the current round alone. However, due to the statistical heterogeneity across clients, the local models from different clients may be greatly diverse, making the obtained global model incapable of maintaining the specific knowledge of each local model. In this paper, we introduce a novel method, FedCDA, which selectively aggregates cross-round local models, decreasing discrepancies between the global model and local models. The principle behind FedCDA is that due to the different global model parameters received in different rounds and the non-convexity of deep neural networks, the local models from each client may converge to different local optima across rounds. Therefore, for each client, we select a local model from its several recent local models obtained in multiple rounds, where the local model is selected by minimizing its divergence from the local models of other clients. This ensures the aggregated global model remains close to all selected local models to maintain their data knowledge. Extensive experiments conducted on various models and datasets reveal our approach outperforms state-of-the-art aggregation methods.

## 1 Introduction

Federated Learning (FL) has emerged as a key framework for training deep neural networks (DNNs) through client collaboration without the need to share original datasets (McMahan et al., 2017b; Wang et al., 2022; Li et al., 2022b). It has been extensively utilized in areas like medical image processing (Liu et al.; Guo et al.; Xu et al.) and recommendation systems (Ramaswamy et al., 2019; Ammad-ud-din et al., 2019). FL is an iterative procedure in which each round involves the local model training across various individual clients, and then aggregating these models centrally on a server (McMahan et al., 2017a).

In this paper, we focus on the *aggregation* of FL, which is the critical step to obtain the global model from multiple local models. The typical aggregation method is FedAvg, which computes the coordinate-wise weighted average of parameters of local models with the weight as the ratio of the data size (McMahan et al., 2017b). Although the implementation of this method is straightforward, some works (Yurochkin et al., 2019a; Li et al., 2022b; Liu et al., 2022; Wang et al., 2020a) consider that the coordinate-wise average will reduce the performance due to the NonIID (i.e., not independently and identically) data among clients. Specifically, they identify that the parameter ordering of different local models may be varied due to the permutation invariance of neural network (NN) parameters. Thus, they propose re-ordering the parameters before applying the weighted average. Another type of work considers that the NonIID data also affects the aggregation weights and they propose adaptively setting the weights using a learnable approach (Li et al., 2023). Although these

---

*Haozhao Wang, Haoran Xu, and Yichen Li contribute equally to this work. Ruixuan Li is the Corresponding author.

methods have achieved great success separately, they mainly aggregate the local models from the current single round, which may limit the improvement of FL performance.

Orthogonal to these works, in this paper, we focus on the aggregation of cross-round local models to further unleash the potential aggregation performance. Intuitively, to acquire the data knowledge of some specific client, it is necessary for the global model to be close to its locally trained model (Kirkpatrick et al., 2017). Nevertheless, the local models of different clients in the same single round may have a large divergence from each other due to the statistical heterogeneity. Thus, as shown in Figure 1(a), the aggregated global model may greatly deviate from these local models. To tackle this challenge, we consider a common fact that each client is usually able to achieve convergence in different rounds after the startup training stage, especially when FL prefers a larger interval for the local training process to save the communication cost (Sun et al., 2023a). In addition, due to receiving different global models in different rounds and the existence of multiple local optima in deep neural networks (Wu et al., 2017; Kawaguchi, 2016; Xie et al., 2021), each client often converges to different models in different rounds, each of which can usually learn local data well, especially when training data using the most advanced optimizer (Loshchilov & Hutter, 2019; Chaudhari et al., 2017). Therefore, a natural idea is that the global model can also fit the local data of some specific client once it approaches any of the different local models in multiple rounds. Motivated by this, the global model can essentially be obtained by aggregating selected local models from different rounds to reduce their divergence and maintain their knowledge.

Based on the above motivation, we propose a novel aggregation method named FedCDA, which selectively aggregates cross-round local models. More specifically, we design a divergence-aware selection strategy that selects local models from multiple rounds with minimum divergence to their aggregated model and only aggregates the selected local models to obtain the global model. In this way, as shown in Figure 1(b), the global model approaches selected local models and thus maintains the data knowledge of clients. Considering the selection problem is a combinatorial optimization problem with a large search

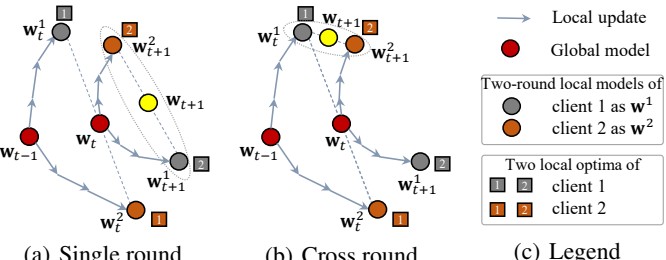

(a) Single round      (b) Cross round      (c) Legend

Figure 1: Comparison of different aggregation methods on a two-client FL. (a) The global model $\mathbf{w}_{t+1}$ is the aggregation of the local model $\mathbf{w}_{t+1}^1$ of client 1 and $\mathbf{w}_{t+1}^2$ of the client 2 in the same $t+1$-th round. (b) $\mathbf{w}_{t+1}$ is the aggregation of the $t$-th round local model $\mathbf{w}_t^1$ and the $t+1$-th round local model $\mathbf{w}_{t+1}^2$. *$\mathbf{w}_{t+1}$ obtained through cross-round aggregation is close to the local optimum 1 of client 1 and the local optimum 2 of client 2, while the single-round aggregation is distant from any local optima.* A practical example is in Appendix A.

space, we further design an approximation version by selecting local models in a batch way to reduce the selection cost. Then, we establish theories to provide a better understanding and guarantee the convergence of our method. We conduct extensive experiments on various datasets, and the results show that FedCDA outperforms state-of-the-art baselines. Our contributions are:

- To the best of our knowledge, this paper is the first to study the aggregation of cross-round local models. We identify that cross-round local models among clients may have a smaller divergence than those in the single round and selectively aggregating them can make the global model approach local models more closely, thus maintaining their local data knowledge.

- We propose a new cross-round aggregation method named FedCDA. It obtains the global model by aggregating local models selected from multiple rounds based on the criterion of the minimum divergence. Besides, we design an approximation strategy to reduce the cost of selection.

- We establish comprehensive theories for our method. Specifically, we provide theoretical insights for understanding our algorithm and show that the approximation selection error is bounded by the convergence of the local model. Besides, we also prove the convergence of our method.

- We conduct extensive experiments over various deep-learning models and datasets. The efficiency superiority of FedCDA is demonstrated by comparing our proposed aggregation method with traditional aggregation methods, which achieves the best performance.

## 2 RELATED WORKS

Many previous methods have been proposed to improve the performance of FL. For example, some works propose regularizing the update of the local model to mitigate the NonIID issue (Li et al., 2020; Sun et al., 2023b). Orthogonal to these works, this paper focuses on the aggregation of local models. Generally, there are three main types of FL aggregation methods.

**Aggregation Weights** One of the typical researches is to determine adaptive aggregation weights (Li et al., 2022a; Rehman et al., 2023). For instance, AUTO-FEDAVG (Xia et al., 2021) tailors weights based on distinct institutional medical datasets to enable personalized medicine, whereas L2C (Li et al., 2022a) identifies similar peers in decentralized FL by adapting weights using local data. While these approaches have proven effective, they primarily emphasize the creation of personalized models for individual clients. In contrast, our work centers on acquiring a global model. Recently, FedLAW (Li et al., 2023) aims to obtain a global model by learning the weights. Nevertheless, all of these methods rely on the proxy dataset in the server while our aggregation method does not.

**Model Fusion** Due to the permutation invariance of neural network parameters, some works consider that the parameters ordering of different local models across clients may be varied especially when their data is NonIID (Yu et al., 2021; Singh & Jaggi, 2020; Li et al., 2022b). In this case, the coordinate-wise average of local models will lead to a mismatch between the same-position parameters of cross-client local models, degrading the performance of the aggregated model. Hence, these works seek to fuse these local models by re-ordering the parameters to match them across clients such as using Hungarian matching algorithm (Wang et al., 2020a), Bayesian approach (Yurochkin et al., 2019b), or a graph matching algorithm (Liu et al., 2022).

**Federated Distillation** Different from the two above types of methods that compute the average of model parameters, federated distillation employs an ensemble distillation computing the average of their logits over the aggregation of local models (Wu & Gong, 2021; Guo et al., 2020; Bistritz et al., 2020; Wang et al., 2023). Notably, Lin et al. (2020); Chen & Chao (2021) initially introduced a technique that harnesses knowledge distillation on the server side. This approach transfers knowledge from multiple local models to the global model using an unlabeled proxy dataset. However, these methods depend on the availability of an auxiliary dataset on the server, which may not be present in real-world scenarios. In response to this limitation, recent studies (Zhu et al., 2021; Zhang et al., 2022; Wang et al., 2023) proposed replacing the proxy dataset with generated data, enabling ensemble federated distillation in a data-free manner. We in this paper focus on the average of model parameters, which is orthogonal to these works.

## 3 SETUP

Federated learning allows $N$ clients with a server to solve the following optimization problem:

$$\min_{\mathbf{w} \in \mathbb{R}^d} F(\mathbf{w}) = \frac{1}{N} \sum_{n=1}^{N} F_n(\mathbf{w}), \quad s.t., F_n(\mathbf{w}) = \mathbb{E}_{\xi \sim \mathcal{D}_n} f_n(\mathbf{w}; \xi) \tag{1}$$

to obtain the global model $\mathbf{w}$. The function $F_n(\mathbf{w}) : \mathbb{R}^d \to \mathbb{R}$ denotes the expected loss over the data distribution of client $n$. $\mathcal{D}_n$ denotes the data distribution of the $n$-th client. $f_n(\mathbf{w}; \xi)$ denotes the loss value with respect to model $\mathbf{w}$ and random data sample $\xi$. Without causing confusion, we use $f_n(\mathbf{w})$ to denote a mini-batch of $f_n(\mathbf{w}; \xi)$ for simplicity. Besides, we make the following assumptions for these objectives which are widely adopted in FL (Dinh et al., 2020; Wang et al., 2020b).

**Assumption 1** *(L-smoothness).* *The objective function $F_n$ is L-smooth with Lipschitz constant $L > 0$, i.e., $\|\nabla F_n(\mathbf{w}) - \nabla F_n(\mathbf{w}')\|_2 \leq L\|\mathbf{w} - \mathbf{w}'\|_2$ for all $\mathbf{w}, \mathbf{w}'$.*

**Assumption 2** *(Bounded Variance).* *For all parameters $\mathbf{w}$, the variance of the local stochastic gradient in each client is bounded by $\sigma_l^2$: $\mathbb{E}(\|\nabla f_n(\mathbf{w}) - \nabla F_n(\mathbf{w})\|^2) \leq \sigma_l^2$. Besides, the global variance of gradients among clients is bounded by $\sigma_g^2$: $\frac{1}{N} \sum_{n=1}^{N} \|\nabla F_n(\mathbf{w}) - \nabla F(\mathbf{w})\|^2 \leq \sigma_g^2$.*

**Assumption 3** *(Bounded Gradient).* *For all parameters $\mathbf{w}$, the stochastic gradient with respect to the loss is bounded by a constant $M$: $\mathbb{E}(\|\nabla f_n(\mathbf{w})\|^2) \leq M^2$.*

## 4 METHODOLOGY

In this part, we will introduce our proposed aggregation method. To minimize the objective (1), we first apply Assumption 1 to each local loss function $F_n(\mathbf{w})$:

$$\min_{\mathbf{w}\in\mathbb{R}^d} F(\mathbf{w}) = \frac{1}{N}\sum_{n=1}^{N}F_n(\mathbf{w}) \leq \frac{1}{N}\sum_{n=1}^{N}\left[F_n(\mathbf{w}^n)+\nabla_{\mathbf{w}^n}F_n(\mathbf{w}^n)(\mathbf{w}-\mathbf{w}^n)+L\|\mathbf{w}-\mathbf{w}^n\|_2^2\right]. \quad (2)$$

Then, we turn to minimize the upper bound of the objective function (1), which corresponds to the right-hand side term of the inequality (2), by aggregating local models $\mathbf{w}^n$ of each client $n$ from multiple rounds. Given the set of recent $K$ local models $\mathcal{W}_t^n = \{\mathbf{w}_{t_1}^n, \ldots, \mathbf{w}_{t_K}^n\}$ of each client $n$ obtained in multiple rounds, the server seeks to solve the following objective:

$$\min_{\mathbf{w}\in\mathbb{R}^d,\mathbf{w}^1\in\mathcal{W}_t^1,\ldots,\mathbf{w}^N\in\mathcal{W}_t^N} \frac{1}{N}\sum_{n=1}^{N}\left[F_n(\mathbf{w}^n) + \nabla_{\mathbf{w}^n}F_n(\mathbf{w}^n)^T(\mathbf{w}-\mathbf{w}^n) + \frac{L}{2}\|\mathbf{w}-\mathbf{w}^n\|_2^2\right]. \quad (3)$$

The problem (3) is strongly convex in terms of $\mathbf{w}$ for any combination of the local models $\mathbf{w}^n$. Therefore, the global model $\mathbf{w}$ has a closed-form solution with respective to the local models $\mathbf{w}^n$:

$$\mathbf{w} = \frac{1}{N}\sum_{n=1}^{N}\mathbf{w}^n - \frac{1}{LN}\sum_{n=1}^{N}\nabla_{\mathbf{w}^n}F_n(\mathbf{w}^n). \quad (4)$$

Given equation (4), the problem (3) is equivalent to a combinatorial optimization problem to select local models $\mathbf{w}^n$. However, solving this problem requires computing the full gradient $\nabla_{\mathbf{w}^n}F_n(\mathbf{w}^n)$ on the local dataset of each client $n$, leading to extra expensive computation and communication cost. Considering the local model $\mathbf{w}^n$ may nearly approach one of the local optima or saddle point $\mathbf{w}^{n,*}$ especially at the end of the FL training stage or when the number of local epochs is large, we take an approximation as $\nabla_{\mathbf{w}^n}F_n(\mathbf{w}^n) \approx 0$. The problem (3) can be re-formulated as:

$$\min_{\mathbf{w}^1\in\mathcal{W}_t^1,\ldots,\mathbf{w}^N\in\mathcal{W}_t^N} \frac{1}{N}\sum_{n=1}^{N}F_n(\mathbf{w}^n) + \frac{L}{2N}\sum_{n=1}^{N}\|\mathbf{w}-\mathbf{w}^n\|_2^2, \quad s.t., \mathbf{w} = \frac{1}{N}\sum_{n=1}^{N}\mathbf{w}^n \quad (5)$$

$$\Leftrightarrow \min_{\mathbf{w}^1\in\mathcal{W}_t^1,\ldots,\mathbf{w}^N\in\mathcal{W}_t^N} \frac{1}{N}\sum_{n=1}^{N}F_n(\mathbf{w}^n) + \frac{L}{2N}\sum_{n=1}^{N}\|\mathbf{w}^n\|_2^2 - \frac{L}{2}\|\mathbf{w}\|_2^2, \quad s.t., \mathbf{w} = \frac{1}{N}\sum_{n=1}^{N}\mathbf{w}^n. \quad (6)$$

*Equation (5) reveals that the criterion for choosing local models can be understood as the selection of cross-round local models that exhibit minimal divergence among each other, i.e., variance* $\frac{1}{N}\sum_{n=1}^{N}\|\mathbf{w}-\mathbf{w}^n\|_2^2$, *particularly when the difference in loss* $F_n(\mathbf{w}^n)$ *tends to be small among clients.* Although solving (5) can obtain the optimal combination of cross-round local models, the computation complexity and memory cost are large. An approach to reducing the computation cost is to utilize the equivalent version of (5), i.e., (6), which is derived using $\frac{1}{n}\sum_{i=1}^{n}(x_i - \bar{x})^2 = \frac{1}{n}\sum_{i=1}^{n}x_i^2 - \bar{x}^2$ with $\bar{x} = \frac{1}{n}\sum_{i=1}^{n}x_i$. In this way, the $l_2$ norm of $\|\mathbf{w}^n\|_2^2$ can be cached once it is computed to avoid repeated computations. Yet, the search space for all combinations is still large. Denoting the model size as $C$, we have the following conclusion.

**Proposition 1** *The computation complexity of solving (6) is $\mathcal{O}(K^N)$ and the memory cost is $KNC$.*

Due to the exponential complexity of computation, directly solving (6) is not affordable even by a cloud for large $K$ and $N$. Therefore, we further propose selecting local models with approximately minimum divergence to reduce the cost. Our strategy includes two steps.

**First, selection for partial clients.** We propose only selecting local models from $P$ clients $n \in \mathcal{P}_t$ that participate in the current round $t$ and fixing the local models of other clients $n \in \mathcal{N} - \mathcal{P}_t$ by using those selected in previous rounds:

$$\min_{\mathbf{w}^n\in\mathcal{W}_t^n, \forall n\in\mathcal{P}_t} \frac{1}{N}\sum_{n\in\mathcal{P}_t}\mathcal{L}_n(\mathbf{w}^n) + \frac{1}{N}\underbrace{\sum_{n\in\mathcal{N}-\mathcal{P}_t}\mathcal{L}_n(\mathbf{w}^n)}_{\text{Fixed in current round}} - \frac{L}{2}\|\mathbf{w}\|_2^2, \quad (7)$$

where the aggregated model $\mathbf{w}$ remains the same as $\mathbf{w} = \frac{1}{N}\sum_{n=1}^{N}\mathbf{w}^n$ and $\mathcal{L}_n(\mathbf{w}^n)$ denotes $\mathcal{L}_n(\mathbf{w}^n) = F_n(\mathbf{w}^n) + \frac{L}{2}\|\mathbf{w}^n\|_2^2$. As the local models of non-participating clients are fixed in the current round, this leads to a great reduction in computational complexity and memory requirements.

**Proposition 2** *The computation complexity of solving (7) is $\mathcal{O}(K^P)$ and the memory cost is $KPC$.*

**Second, batch-based selection.** To further reduce the computation complexity, we propose selecting local models in a stochastic greedy manner. Specifically, we randomly group participated clients into $B$ equal-size batches $\mathcal{P}_t = \mathcal{P}_t^1 \cup \cdots \cup \mathcal{P}_t^B$ and select local models for these clients batch by batch. When selecting local models for clients in the $b$-th batch, the local models of clients in batches 1 to $b-1$ are fixed and in batches $b+1$ to $P$ are excluded, and the objective is:

$$\min_{\mathbf{w}^n \in \mathcal{W}_t^n, \forall n \in \mathcal{P}_t^b} \frac{1}{N-P+\frac{bP}{B}} \left( \sum_{n \in \mathcal{P}_t^b} \mathcal{L}_n(\mathbf{w}^n) + \underbrace{\sum_{n \in (\mathcal{N}-\mathcal{P}_t) \cup \mathcal{P}_t^1 \cup \cdots \cup \mathcal{P}_t^{b-1}} \mathcal{L}_n(\mathbf{w}^n)}_{\text{Fixed in the } b\text{-th subset selection}} \right) - \frac{L}{2}\|\mathbf{w}\|_2^2, \quad (8)$$

where the model $\mathbf{w}$ is aggregated by computing the average of local models of non-participated clients and the 1-st to $b$-th batch of participated clients, i.e., $\mathbf{w} = \frac{1}{N-P+\frac{bP}{B}} \sum_{n \in (\mathcal{N}-\mathcal{P}_t) \cup \mathcal{P}_t^1 \cup \cdots \cup \mathcal{P}_t^b} \mathbf{w}^n$. This can also be viewed as selecting local models that are close to that of clients participating in previous rounds and thus maintaining the memory of their data. The complexity of the computation is further reduced.

**Proposition 3** *The computation complexity of solving (8) is $\mathcal{O}(BK^{\frac{P}{B}})$ and the memory cost is $KPC$.*

While the computational complexity remains exponential, we retain the flexibility to manually adjust the value of $B$ for control. In practice, we can maintain $\frac{P}{B}$ as a constant, effectively reducing the complexity to an acceptable level. In an extreme scenario, we can set $B = P$, resulting in linear complexity with respect to the value of $K$. Our experiments have shown that even a small value of $K$, such as $K = 3$, produces satisfactory performance, rendering the computational complexity acceptable for practical applications. Additionally, the memory cost $KPC$ is also manageable when $K$ is small, because the number of sampled clients $P$ is usually a small ratio of the total clients. The local models of non-participated clients can be stored on the disk which has sufficient storage space. For example, a 1TB hard drive can store approximately $20,000$ copies of ResNet-18, which is widely adopted on the edge. Given that even a mobile phone is equipped with 1 TB storage, we believe that the cost is within the budget of the aggregation node which is typically hosted by a cloud.

As compared to other aggregation methods like weight setting (Li et al., 2023) or ensemble distillation Lin et al. (2020); Chen & Chao (2021), our approach has a distinct advantage. We do not depend on an additional public dataset, which can be challenging to acquire due to the requirement for a similar distribution as the global dataset. Moreover, our method does not introduce higher computational complexity compared to existing methods. Many existing aggregation methods involve performing gradient descent (Li et al., 2023; Chen & Chao, 2021) or solve maximum bipartite matching problems (Wang et al., 2020a), which can be computationally intensive.

### 4.1 FEDCDA ALGORITHM

The complete procedure of our method is given in Algorithm 1 by assuming the base algorithm is FedAvg (McMahan et al., 2017a). `FedCDA` differs from FedAvg primarily in lines 9 and 10 of its implementation. When it receives local models from a subset of clients, the server updates its cached local models. This update involves replacing the oldest round's local model with the most recently received one, as indicated in line 9. After this update, the server selects local models for aggregation by solving the problem (8) in line 10. Finally, the global model is obtained by averaging both selected and fixed local models to retain knowledge contributed by all clients.

In practice, *we usually apply* `FedCDA` *after a warmup training stage* using FedAvg or other baselines to ensure that the local models can approach the convergence during the local training process. It is also worthwhile to note that most existing methods usually employ improved techniques over the step of line 11, e.g., re-setting aggregation weights (Li et al., 2023) or using ensemble distillation (Lin et al., 2020; Chen & Chao, 2021), which are orthogonal to us.

## 5 THEORETICAL ANALYSIS

In this section, we provide theories for better understanding the principles and bounding the error of the proposed algorithm. We first prove that the selection error of using the approximated objective

---

**Algorithm 1** FedCDA Algorithm

---

**Input:** Number of cached local models $K$, number of subsets $B$, learning rate $\eta$, number of sampling clients $P$, and total communication rounds $T$.

**Output:** Converged global model $\mathbf{w}$.

 1: Initialize the model parameter $\mathbf{w}_0$;
 2: Distribute $\mathbf{w}_0$ to all clients;
 3: **for** each communication round $t \in \{1, 2, ..., T\}$ **do**
 4:     Randomly select a set of clients $\mathcal{P}_t$;
 5:     **for** each selected client $n \in \mathcal{P}_t$ **in parallel do**
 6:         Initialize the local model with the received global model: $\mathbf{w}^n = \mathbf{w}_t$;
 7:         Solve the local problem by updating $\mathbf{w}_n$ for $E$ local mini-batch SGD steps and accumulate the local loss $F_n(\mathbf{w}^n)$ in the last local epoch: $\mathbf{w}^n = \mathbf{w}^n - \eta \nabla_{\mathbf{w}^n} f_n(\mathbf{w}^n)$;
 8:     Update the cached set $\mathcal{W}_t^n$ for $n \in \mathcal{P}_t$ by replacing the oldest model with received $\mathbf{w}^n$;
 9:     Select local models $\mathbf{w}^n$ for each client $n \in \mathcal{P}_t$ by solving the problem (8);
 10:     Aggregate both selected and fixed local models to obtain: $\mathbf{w}_{t+1} = \frac{1}{N} \sum_{n=1}^{N} \mathbf{w}^n$;
     **return** global model $\mathbf{w}_T$

---

(5) to the exact objective (3) is bounded by the convergence degree of local models. Then, we present the benefits of FedCDA on idealized cases and conditions. Finally, we establish the convergence theories for our algorithm.

**Theorem 1** (*Approximation Selection Error*) *Define the local optima closest to $\mathbf{w}_t^n$ as $\mathbf{w}_t^{n,*}$ and the maximum distance between any two local optima that are close to cached local models across clients and rounds as $D$, i.e., $D = max_{n \in [N], n' \in [N], n \neq n', i \in [K], i' \in [K]}(\|\mathbf{w}_{t_i}^{n,*} - \mathbf{w}_{t_{i'}'}^{n',*}\|)$. If the distance between the local model $\mathbf{w}_t^n$ and its approximated critical point $\mathbf{w}_t^{n,*}$ is limited by a constant $\epsilon > 0$, i.e., $\|\mathbf{w}_t^n - \mathbf{w}_t^{n,*}\| \leq \epsilon$, then the disparity in the global loss between aggregating local models selected using (5) and (3) is constrained by $\varepsilon \leq 4L\epsilon^2 + 2LD\epsilon$.*

The proof can be found in Appendix B.1. The theorem indicates that the approximation error of using (5) to (3) becomes smaller when local models are convergent. It implicitly reveals that our algorithm may obtain a better global model when the local models approach convergence, i.e., with large local iterations or large warmup rounds, which are verified by our experimental results in Figure 2(c) and Figure 6.2. Further, we seek to show that minimizing (3) leads to a lower global loss than naively aggregating the local models in the newest current round. We define the divergence among local models $\mathbf{w}^n, \forall n = 1 \ldots, N$ as $\text{Var}(\mathbf{w}^n) = \frac{1}{N} \sum_{n=1}^{N} \|\frac{1}{N} \sum_{n=1}^{N} \mathbf{w}^n - \mathbf{w}^n\|_2^2$. We denote $\mathbf{w}_{t_*}, \mathbf{w}_{t_*}^n, n = 1, \ldots, N$ as the solution of objective (3). Similarly, we denote $\mathbf{w}_t$ as the $t$-th round global model aggregated from all $t$-th round local models $\mathbf{w}_t^n, \mathbf{w}_{t_*}^n, n = 1, \ldots, N$. Subsequently, we demonstrate that the global loss $F(\mathbf{w}_{t_*})$ can be assured to be lower than $F(\mathbf{w}_t)$ under the condition that the divergence $\text{Var}(\mathbf{w}_{t_*}^n)$ is less than $\text{Var}(\mathbf{w}_t^n)$ by a certain value.

**Theorem 2** (*Impact of Divergence of Local Optima*) *Let the definition of the local optima $\mathbf{w}_t^{n,*}$ and distance $\epsilon$ be the same as Theorem 1. Consider the loss function $F_n(\mathbf{w})$ is strongly convex with a parameter $\mu$ within the region spanning from the local optima $\mathbf{w}_t^{n,*}$ to the global model $\mathbf{w}_t$ and the local loss achieves equivalent values on local optima in different rounds, i.e., $F_n(\mathbf{w}_t^{n,*}) = F_n(\mathbf{w}_{t'}^{n,*})$. If the divergence among selected local models is small enough, i.e., satisfying $\text{Var}(\mathbf{w}_{t_*}^{n,*}) \leq \frac{\mu}{L}\text{Var}(\mathbf{w}_t^{n,*}) - (\frac{\mu}{L} + 1)\epsilon^2$, then the global loss of using selected global model $\mathbf{w}_{t_*}$ is smaller than that of using $t$-th round global model $\mathbf{w}_t$, i.e., $F(\mathbf{w}_{t_*}) \leq F(\mathbf{w}_t)$.*

The proof can be found in Appendix B.2. Although the conditions of Theorem 2 may be idealized in practical settings, it provides some insights for understanding our method. Smaller divergence among local models leads to a smaller loss of the aggregated model. An ideal case is that the divergence is reduced to 0 where the local optima of all local models across clients are the same. In fact, such an ideal case can widely exist in overparameterized deep neural networks, where a large model may achieve 0 loss in the local dataset of each client and hence is the local optima of all clients. Therefore, our method may prefer large models. The experimental results in Table 1 also verify our statement, where the improvement is higher for the larger models. Although our

motivation mainly comes from the non-convex functions where there are multiple local optima, our algorithm is also applicable to convex cases. More discussions can be found in Appendix B.3. Finally, we present the convergence of our algorithm. Noting that even though our algorithm does not achieve faster theoretical convergence using existing optimization analytical tools, our algorithm demonstrates great empirical benefits.

**Theorem 3** *(**Convergence on Non-convex Functions**) Consider problem (1) under Assumption 1,2, and 3. If the learning rate $\eta$ satisfies $0 < \eta \leq \frac{1}{LE}$, then the global model $\mathbf{w}_{t_*}$ solved by (3) achieves asymptotic convergence, i.e., $\frac{1}{T} \sum_{t=1}^{T} \|\nabla F(\mathbf{w}_{t_*})\|_2^2 = \mathcal{O}(\frac{1}{\sqrt{T}})$.*

*Ideas of Proof*: Our proof mainly includes two parts. First, we prove that the difference between the loss of the global model $\mathbf{w}_{t_*}$ obtained by (3) and that of the reference global model $\mathbf{w}_t$ obtained by aggregating the newest local models is bounded. Then, we prove that the loss of the global model $\mathbf{w}_t$ achieves convergence, which in turn indicates the convergence of the global model $\mathbf{w}_{t_*}$. Detailed derivations are deferred to Appendix B.4.

## 6 EVALUATION

### 6.1 EXPERIMENTAL SETUP

**Datasets and Models:** We consider three popular datasets in experiments: Fashion-MNIST (Xiao et al., 2017), CIFAR-10 (Krizhevsky et al., 2009) and CIFAR-100 (Krizhevsky et al., 2009), which contains 10, 10, 100 classes respectively. For CIFAR-10 and CIFAR-100 datasets, we use ResNet-18 (He et al., 2016) as the backbone to train and test the performance while for Fashion-MNIST we use a simple CNN instead. The simple CNN has two 5x5 convolution layers (the first with 32 channels, the second with 64, each followed with 2x2 max pooling), a fully connected layer with 512 units and ReLu activation, and a final fully output connected layer.

**Data Partition:** To evaluate the performance of our work in a heterogeneous scenario, we specify two Non-IID data partition methods called Shards (McMahan et al., 2017a) and Dirichlet (Lin et al., 2020). In the Shards setting, the sorted samples are shuffled into $N * S$ shards, and assigned to $N$ clients randomly. Each client owns an equal number of pieces. In the second setting, data distribution over clients satisfies the Dirichlet distribution by using $\alpha$ to characterize the degree of heterogeneity. We set $\alpha$ of Dirichlet: $\{0.1, 0.3, 0.5\}$ and shards for each client: $\{2, 4, 8\}$.

**Baselines:** Beside of FedAvg (McMahan et al., 2017a), we also compare against various types of efficient federated learning approaches with the proposed method in our experiments. The first main type includes typical non-aggregation methods that speed FL in the local process or tuning learning rate, including FedProx (Li et al., 2020), FedExP (Jhunjhunwala et al., 2023), and FedSAM (Qu et al., 2022). The methods of the second type can be divided into three main representative aggregation categories: ensemble distillation including FedDF (Lin et al., 2020) and FedGEN (Chen & Chao, 2021); model fusion including FedMA (Wang et al., 2020a) and GAMF (Liu et al., 2022); weights setting including FedLAW (Li et al., 2023).

**Implementation:** We implement the whole experiment in a simulation environment based on PyTorch 2.0 and 8 NVIDIA GeForce RTX 3090 GPUs. We use 20 clients in total and randomly choose 20% each round for local training. We set the local epoch to 20, batch size to 64, and learning rate to $1e-3$. We employ SGD optimizer with momentum of $1e-4$ and weight decay of $1e-5$ for all methods and datasets. At the same time, we set the number of global communication rounds to 200. Each experiment setting is run twice and we take each run's final 10 rounds' accuracy and calculate the average value and standard variance. For our method, we also need to set the memory size of the client $K$ to 3, batch number $B$ to 3, and the number of warmup rounds to 50. Besides, we simply assume $L = 1$ for all clients to save the computation cost.

### 6.2 EXPERIMENT RESULTS

**Performance Comparison.** We report the comparison results with other baselines in Table 1. The results with a broader range of hyperparameters can be found in Appendix D. In order to demonstrate the generalization of our method, we compare them on two different Non-IID settings, Shards and Dirichlet distribution. We apply different data distributions on different datasets. We can see that our proposed `FedCDA` achieves the best performance on almost all settings. It demonstrates the effectiveness and benefit of cross-round divergence-aware aggregation. Specifically, on relatively

Table 1: The comparison of test accuracy of different methods. The best results are **bolded**.

| Method | Fashion-MNIST(%) | | | CIFAR-10(%) | | | CIFAR-100(%) | | |
|---|---|---|---|---|---|---|---|---|---|
| *Shards (S)* | 2 | 4 | 8 | 2 | 4 | 8 | 2 | 4 | 8 |
| FedAvg | $64.69_{\pm5.62}$ | $74.78_{\pm4.55}$ | $76.81_{\pm3.33}$ | $28.10_{\pm3.96}$ | $59.83_{\pm2.94}$ | $70.87_{\pm1.91}$ | $11.86_{\pm1.19}$ | $15.87_{\pm1.00}$ | $21.91_{\pm0.55}$ |
| FedProx | $64.21_{\pm4.11}$ | $70.76_{\pm3.89}$ | $72.19_{\pm4.16}$ | $26.39_{\pm4.16}$ | $53.03_{\pm2.29}$ | $70.91_{\pm1.87}$ | $10.87_{\pm0.58}$ | $15.37_{\pm0.46}$ | $24.16_{\pm0.33}$ |
| FedExP | $65.24_{\pm3.47}$ | $69.31_{\pm4.62}$ | $76.66_{\pm5.04}$ | $26.84_{\pm4.75}$ | $59.31_{\pm3.61}$ | $69.53_{\pm1.94}$ | $11.59_{\pm0.81}$ | $16.47_{\pm0.99}$ | $23.58_{\pm1.36}$ |
| FedSAM | $59.28_{\pm0.15}$ | $75.19_{\pm0.10}$ | $76.07_{\pm0.09}$ | $29.31_{\pm0.32}$ | $57.12_{\pm0.08}$ | $61.56_{\pm0.31}$ | $11.19_{\pm0.16}$ | $15.95_{\pm0.15}$ | $22.44_{\pm0.16}$ |
| FedDF | $64.72_{\pm2.11}$ | $74.16_{\pm1.52}$ | $\mathbf{85.51_{\pm0.95}}$ | $32.37_{\pm2.39}$ | $60.08_{\pm5.67}$ | $71.52_{\pm2.67}$ | $11.63_{\pm0.67}$ | $17.13_{\pm1.12}$ | $25.84_{\pm1.02}$ |
| FedGEN | $63.50_{\pm3.27}$ | $69.42_{\pm4.09}$ | $80.17_{\pm4.71}$ | $27.21_{\pm3.12}$ | $57.16_{\pm2.71}$ | $68.93_{\pm1.75}$ | $10.07_{\pm0.19}$ | $15.26_{\pm0.29}$ | $21.49_{\pm0.17}$ |
| FedMA | $64.71_{\pm4.92}$ | $74.98_{\pm5.03}$ | $77.13_{\pm4.10}$ | $28.61_{\pm1.39}$ | $59.97_{\pm0.96}$ | $70.91_{\pm1.02}$ | $11.89_{\pm0.57}$ | $15.90_{\pm0.92}$ | $22.02_{\pm0.82}$ |
| GAMF | $64.97_{\pm3.93}$ | $75.21_{\pm4.05}$ | $77.34_{\pm3.78}$ | $28.92_{\pm1.52}$ | $60.23_{\pm1.93}$ | $71.44_{\pm1.75}$ | $11.98_{\pm0.99}$ | $16.76_{\pm0.77}$ | $24.15_{\pm0.49}$ |
| FedLAW | $60.34_{\pm4.39}$ | $73.93_{\pm4.91}$ | $77.53_{\pm3.52}$ | $26.32_{\pm2.80}$ | $46.81_{\pm3.61}$ | $61.08_{\pm2.61}$ | $11.57_{\pm1.61}$ | $15.99_{\pm0.49}$ | $22.37_{\pm0.68}$ |
| Ours | $\mathbf{66.30_{\pm0.07}}$ | $\mathbf{76.59_{\pm0.25}}$ | $78.99_{\pm0.13}$ | $\mathbf{34.97_{\pm0.31}}$ | $\mathbf{62.81_{\pm0.28}}$ | $\mathbf{72.04_{\pm0.23}}$ | $\mathbf{12.20_{\pm0.13}}$ | $\mathbf{19.98_{\pm0.25}}$ | $\mathbf{28.16_{\pm0.30}}$ |
| *Dirichlet ($\alpha$)* | 0.1 | 0.3 | 0.5 | 0.1 | 0.3 | 0.5 | 0.1 | 0.3 | 0.5 |
| FedAvg | $71.81_{\pm5.61}$ | $75.97_{\pm3.21}$ | $79.73_{\pm1.94}$ | $50.43_{\pm1.68}$ | $61.11_{\pm2.68}$ | $67.37_{\pm1.69}$ | $30.13_{\pm0.70}$ | $35.73_{\pm0.56}$ | $38.86_{\pm0.35}$ |
| FedProx | $70.44_{\pm3.87}$ | $72.17_{\pm4.10}$ | $75.24_{\pm2.19}$ | $38.98_{\pm5.91}$ | $61.64_{\pm1.92}$ | $70.16_{\pm2.03}$ | $32.96_{\pm1.18}$ | $40.81_{\pm0.41}$ | $42.53_{\pm0.48}$ |
| FedExP | $73.42_{\pm4.22}$ | $76.57_{\pm3.39}$ | $80.22_{\pm3.78}$ | $60.63_{\pm4.32}$ | $70.22_{\pm2.40}$ | $74.37_{\pm1.91}$ | $36.76_{\pm1.18}$ | $44.18_{\pm0.53}$ | $47.80_{\pm0.58}$ |
| FedSAM | $71.65_{\pm0.07}$ | $75.91_{\pm0.06}$ | $77.67_{\pm0.10}$ | $49.96_{\pm0.20}$ | $59.53_{\pm0.19}$ | $64.54_{\pm0.21}$ | $21.54_{\pm0.12}$ | $24.72_{\pm0.18}$ | $28.59_{\pm0.19}$ |
| FedDF | $\mathbf{80.03_{\pm1.04}}$ | $84.42_{\pm0.62}$ | $86.84_{\pm1.93}$ | $54.28_{\pm2.39}$ | $69.85_{\pm5.67}$ | $73.76_{\pm2.67}$ | $34.76_{\pm0.67}$ | $39.42_{\pm1.12}$ | $42.31_{\pm1.02}$ |
| FedGEN | $73.02_{\pm1.87}$ | $77.48_{\pm3.50}$ | $81.76_{\pm4.21}$ | $47.09_{\pm3.12}$ | $64.90_{\pm2.71}$ | $68.74_{\pm1.75}$ | $29.02_{\pm0.19}$ | $38.54_{\pm0.29}$ | $40.81_{\pm0.17}$ |
| FedMA | $71.87_{\pm4.28}$ | $75.89_{\pm4.15}$ | $80.12_{\pm3.23}$ | $49.98_{\pm2.01}$ | $61.32_{\pm2.17}$ | $68.42_{\pm1.95}$ | $30.02_{\pm0.58}$ | $36.21_{\pm0.83}$ | $39.55_{\pm0.52}$ |
| GAMF | $72.11_{\pm5.16}$ | $76.24_{\pm3.67}$ | $80.55_{\pm2.06}$ | $51.21_{\pm1.37}$ | $63.45_{\pm1.03}$ | $70.14_{\pm1.81}$ | $31.12_{\pm0.69}$ | $37.26_{\pm0.78}$ | $41.25_{\pm0.74}$ |
| FedLAW | $71.93_{\pm8.23}$ | $76.88_{\pm2.80}$ | $79.98_{\pm1.09}$ | $48.91_{\pm3.59}$ | $61.50_{\pm2.29}$ | $67.08_{\pm1.75}$ | $32.01_{\pm2.61}$ | $38.80_{\pm2.20}$ | $40.11_{\pm1.17}$ |
| Ours | $78.63_{\pm0.14}$ | $\mathbf{84.67_{\pm0.12}}$ | $\mathbf{87.01_{\pm0.08}}$ | $\mathbf{62.46_{\pm0.22}}$ | $\mathbf{70.27_{\pm0.29}}$ | $\mathbf{74.96_{\pm0.17}}$ | $\mathbf{39.38_{\pm0.25}}$ | $\mathbf{45.86_{\pm0.22}}$ | $\mathbf{49.31_{\pm0.22}}$ |

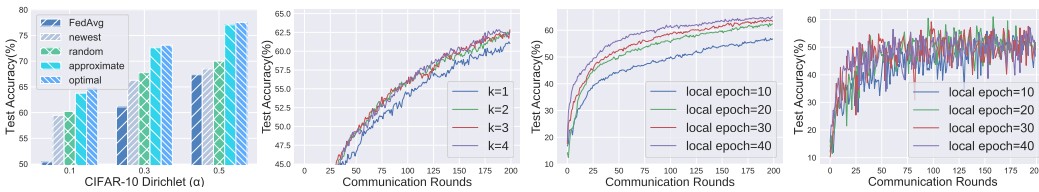

(a) Aggregation strategies     (b) Memory size $K$     (c) Local ep.s (`FedCDA`)     (d) Local ep.s (FedAvg)

Figure 2: (a) shows the effect of different aggregation strategies. (b) shows the impact of the memory size $K$ on `FedCDA`. (c) and (d) present the impact of local epochs (ep.s) on `FedCDA` and FedAvg.

larger datasets such as CIFAR-10 Dirichlet 0.1, `FedCDA` with ResNet-18 achieves 62.46% accuracy whereas the best baseline method FedExP achieves 60.63% accuracy. In addition, `FedCDA` with simple CNN also makes improvements on relatively smaller datasets, although the improvement is less than in large models. At the same time, we can also see that the results of our method on relatively small datasets and simple CNN are not the best, which may be because the features of models with different rounds are more similar on small datasets and simple models, and can not provide more aggregation features to accelerate convergence. In conclusion, we can notice our `FedCDA` makes more improvements on the large model and complex datasets.

**More Comparison Results with Different Hyper-parameters.** To compare with baselines in a comprehensive way, we further conduct experiments on different hyper-parameters. The number of clients is 100 with the sample ratio being 10%. The learning rate is set to be 0.1 with the weight decay being 1e-3, and the number of local epochs is 5. The local optimizer is SGD without momentum. The experiment is conducted by running the ResNet18 on the CIFAR-100 dataset. The results are

Table 2: Results of FL with 100 clients.

| Method | Dir(0.1) | Dir(0.3) | Dir(0.5) |
|---|---|---|---|
| FedAvg | $38.89\%_{\pm0.85\%}$ | $40.38\%_{\pm0.55\%}$ | $42.23\%_{\pm0.30\%}$ |
| FedProx | $39.86\%_{\pm0.45\%}$ | $39.48\%_{\pm0.37\%}$ | $40.18\%_{\pm0.46\%}$ |
| FedExP | $38.04\%_{\pm3.37\%}$ | $44.10\%_{\pm1.69\%}$ | $41.79\%_{\pm1.62\%}$ |
| FedSAM | $16.35\%_{\pm0.25\%}$ | $20.53\%_{\pm0.25\%}$ | $25.70\%_{\pm0.28\%}$ |
| FedDF | $41.04\%_{\pm0.57\%}$ | $47.06\%_{\pm0.74\%}$ | $47.63\%_{\pm0.53\%}$ |
| FedGEN | $39.91\%_{\pm1.72\%}$ | $41.65\%_{\pm1.35\%}$ | $43.39\%_{\pm1.08\%}$ |
| FedMA | $39.12\%_{\pm0.52\%}$ | $40.42\%_{\pm0.61\%}$ | $42.89\%_{\pm0.21\%}$ |
| GAMF | $39.89\%_{\pm0.67\%}$ | $40.98\%_{\pm0.32\%}$ | $43.25\%_{\pm0.34\%}$ |
| FedLAW | $40.88\%_{\pm0.66\%}$ | $41.77\%_{\pm0.78\%}$ | $41.89\%_{\pm0.33\%}$ |
| Ours | $\mathbf{47.38\%_{\pm0.23\%}}$ | $\mathbf{49.96\%_{\pm0.21\%}}$ | $\mathbf{50.04\%_{\pm0.19\%}}$ |

shown in Table 2. As can be seen, our method still performs the best. Specifically, when the data is the most heterogeneous, i.e., with Dirichlet $\alpha = 0.1$, `FedCDA` achieves the accuracy of 47.38% which outperforms the best baseline method FedDF by 6.34%.

**Different Aggregation Strategies.** We compare five aggregation strategies on the CIFAR-10 datasets. Because the *optimal* selection for updates method is an exponential method, we only sample 10 clients in each round, where the sample ratio is 0.3 and the client memory size is $K = 3$. As shown in Figure 2(a), we compare FedAvg with different aggregation strategies in our method. The

clients in FedAvg do not require memory. The *newest* strategy is that only the newest local model of each client is aggregated during the server aggregation phase. The *random* strategy is that during the server aggregation phase, we randomly select a local model from multiple rounds to aggregate. Finally, the *approximate* strategy is as 8 shown above and the optimal one is as 3. We can find that the approximate and optimal strategies have huge performance improvement over FedAvg, newest and random strategies with ResNet-18 on CIFAR-10 Dirichlet 0.1, 0.3 and 0.5. At its peak, there is an almost 10% increase over FedAvg. We can also see the performance of approximate performance is about the same as the optimal one, but the convergence time of the former is much smaller than that of the latter. In fact, the former is actually a greedy algorithmic approximation of the latter, so the computation of the solution is greatly reduced. We also compare the average polymerization time of each round of these aggregation strategies. Details are in the Appendix C.

**Hyperparameters Sensitivity.** As shown in the following figure 2(b) , We compare the test accuracy of client memory size $K$ for 1, 2, 3, 4 on CIFAR-10 Dirichlet 0.1. As $K$ increases, the final test accuracy increases which confirms our theory. The increase of $K$ value gives cross-round polymerization more choices and possibilities.

**Comparison with Different Epochs for `FedCDA` and FedAvg.** By comparing the results in Figure 2(c) and Figure 2(d), from the vertical perspective, our `FedCDA` eventually converges with increasing test accuracy with more local epochs while the final convergence accuracy of Fedavg remains roughly unchanged. *This proves that our algorithm can tolerate large local interactions to save communication cost.* Horizontally, `FedCDA` converges rapidly and stably, whereas the convergence curve of FedAvg is very oscillatory. The reason is that our method excludes the negative impact of sampling clients while FedAvg cannot. Therefore, *the convergence of our method is more stable than FedAvg.*

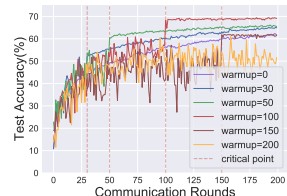

Figure 3: Impact of Batch

**Effect of Batch Number.** As we can see in Figure 6.2, different batch numbers have little effect on the final precision result. Our experiment setup 50 clients and 10 sample clients on the CIFAR-10 Dirichlet 0.1. We compare the results for batch $B = 1, 2, 3, 4, 5, 6, 7$. The results show that the approximation selection can keep the accuracy closer to the optimal selection.

**Warmup Analysis.** The experiments of Figure 6.2 are conducted on settings of CIFAR-10, 20 clients, and the sample ratio 0.2. `FedCDA` with no warmup rounds is worse than some with warmup rounds. This is because the local models in the early rounds can not approach convergence. *The combination with not-well-converged local models in old rounds may prevent the training of the global model.* Therefore, it is similar to FedAvg in the startup training stages. Its advantages gradually exhibit with the training proceeds and outperforms FedAvg (warmup=200). Yet, there is a threshold for raising warmup rounds. Specifically, we can see that FL with 150 warmup rounds has worse performance than 100 warmup rounds. The principle behind it is that the local models and the global model have approached convergence in the final training stage. *The difference between local models in different rounds is greatly reduced and thus the combination of them gains little benefits.*

Figure 4: Impact of Warmup

## 7 CONCLUSION

This paper targets aggregation in federated learning, addressing the issue that traditional single-round methods may not preserve locally learned knowledge due to statistical heterogeneity. Recognizing clients' convergence post-startup stage and local models' consistent data fitting across rounds, we propose `FedCDA`- a new method that selectively aggregates cross-round models with minimum divergence. To enhance efficiency, we introduce an approximation selection algorithm. Theoretical convergence is proven and empirical results show our method outperforms state-of-the-art baselines.

This paper addresses the ideal scenario where smoothness value is equal among clients. The goal is to improve our method for cases with varying smoothness by refining objectives and incorporating sharpness in cross-round aggregation, as we currently treat all local models equally without considering the sharpness of their local optima, despite flatter optima often correlating with better generalization.

ACKNOWLEDGMENTS

The research is supported under the National Key R&D Program of China (2022ZD0160201) and the RIE2020 Industry Alignment Fund - Industry Collaboration Projects (IAF-ICP) Funding Initiative, as well as cash and in-kind contributions from the industry partner(s). This work is supported by National Natural Science Foundation of China under grants U1836204, U1936108, 62206102, and Science and Technology Support Program of Hubei Province under grant 2022BAA046.

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
