# OpenReview forum: "FedCDA: Federated Learning with Cross-rounds Divergence-aware Aggregation"
_ICLR.cc/2024/Conference — ICLR 2024 poster_

### Official Review · Reviewer_rxU6 · 2023-10-30

**Soundness:** 2 fair
**Presentation:** 1 poor
**Contribution:** 3 good
**Rating:** 6
**Confidence:** 4

**Summary:**

Unlike existing federated learning methods that aggregate the weights of local models of clients participating in the current round, this paper proposes an aggregation method that considers local models from recent rounds for the first time. This idea uses the fact that local models in different rounds within a client converge to different local optimal points after the warm up stage, as clients perform local training of multiple epochs to reduce communication costs. This paper also proposes several tricks to simplify the computation of which local model to select and aggregate among local models from multiple rounds. In addition to the theoretical background of the proposed method, experiments show that it has better performance compared to other aggregation methods.

**Strengths:**

The idea of selectively aggregating local models from several recent rounds is novel. It seems to have successfully overcome the limitations of existing aggregation methods by taking advantage of the fact that deep learning models have multiple local optimal points.

The feasibility and reasoning of the proposed method were confirmed through several theoretical backgrounds. Also, several tricks were presented to reduce the computational complexity of the proposed method.

**Weaknesses:**

The effectiveness of the proposed method was not sufficiently proven in the evaluation section, so doubts remain about the proposed method.

Considering that the proposed method is about a new aggregation method, 20 clients and a participation ratio of 20% are too unrealistic.

As shown in figure 2 (c), the proposed method is sensitive to local epoch hyperparameter because it assumes that local models converge. However, most experiments are conducted at a local epoch of 20, and such a large local epoch makes comparisons with baselines unfair. It is of course an advantage that the performance of the proposed method is good even when the local epoch is large, but this does not mean that simply increasing the local epoch is always good.

In addition to the local epoch, I believe that other hyperparameters used in the experiment are not suitable for comparison with baselines.

For example, the learning rate was set to 1e-3, which helps the local model to converge quickly with a large local epoch. However, it is unclear whether such a small learning rate and large local epoch are suitable for other baselines.

Based on the code included in the supplementary material, I reproduced the FedAvg experiment of CIFAR100 - Dirichlet (0.5) in Table 1. Using the same hyperparameter presented in the paper, FedAvg showed an accuracy of about 38%, which is almost similar to 38.86% recorded in Table 1. However, when lr is 0.1, weight decay is 1e-3, local epoch is 5 and the same experiment was performed with SGD without momentum, the accuracy was about 49%, which is almost similar to the proposed method (49.31%). This does not mean that the proposed method shows similar performance to FedAvg, but I just think that the hyperparmaters should have been chosen a little more carefully. (If possible, can you run the experiment again with the hyperparameters mentioned above?)


The overall writing quality of the paper is not good. There are many incorrect parts in the notation related to formulas and figures in the paper, and many sentences require revision.

**Questions:**

Please refer to weaknesses.

---

> ### Author Response · Authors · 2023-11-16
> **Responses to weakness 1**
>
> ```
> W1. The effectiveness of the proposed method was not sufficiently proven in the evaluation section, so doubts remain about the proposed method.
> Considering that the proposed method is about a new aggregation method, 20 clients and a participation ratio of 20% are too unrealistic.
> ```
>
> **R1**. Thanks for the comments which help improve the comprehensiveness of our evaluations. As we know, there are many papers of which the experiments are conducted with 20 clients [1,2,3].  To avoid the sensitivity of our method over the number of clients, we conduct extensive experiments over **100** clients with a **participation ratio of 10%** and compare our method with FedAvg. The dataset is cifar-100, the learning rate is 0.1, the weight decay is 1e-3, the local epoch is 10, and the optimizer is SGD without momentum. The results are presented below:
>
> | Method | Dir(0.1) | Dir(0.3) | Dir(0.5) |
> |--------|---------|---------|---------|
> | FedAvg | 38.89%(+-0.85%) | 40.38%(+-0.55%) | 42.23%(+-0.30%) |
> | FedProx | 39.86%(+-0.45%) | 39.48%(+-0.37%) | 40.18%(+-0.46%) |
> | FedExP | 38.04%(+-3.37%) | 44.10%(+-1.69%) | 41.79%(+-1.62%) |
> | FedSAM | 16.35%(+-0.25%) | 20.53%(+-0.25%) | 25.70%(+-0.28%) |
> | FedDF | 41.04%(+-0.57%) | 47.06%(+-0.74%) | 47.63%(+-0.53%) |
> | FedGEN | 39.91%(+-1.72%) | 41.65%(+-1.35%) | 43.39%(+-1.08%) |
> | FedMA | 39.12%(+-0.52%) | 40.42%(+-0.61%) | 42.89(+-0.21%) |
> | GAMF | 39.89%(+-0.67%) | 40.98%(+-0.32%) | 43.25(+-0.34%) |
> | FedLAW | 40.88%(+-0.66%) | 41.77%(+-0.78%) | 41.89(+-0.33%) |
> | FedCDA | **47.38%** (+-0.23%) | **49.96%** (+-0.21%) | **51.04%** (+-0.19%) |
>
>
>
> Our method can still achieve higher accuracy than FedAvg.
>
> [1] Zexi Li, Tao Lin, Xinyi Shang, Chao Wu: Revisiting Weighted Aggregation in Federated Learning with Neural Networks. ICML 2023
>
> [2] Zhuangdi Zhu, Junyuan Hong, Jiayu Zhou: Data-Free Knowledge Distillation for Heterogeneous Federated Learning. ICML 2021.
>
> [3] Tao Lin, Lingjing Kong, Sebastian U. Stich, Martin Jaggi: Ensemble Distillation for Robust Model Fusion in Federated Learning. NeurIPS 2020

---

> ### Author Response · Authors · 2023-11-16
> **Responses to weakness 2 and 3**
>
> ```
> W2. As shown in figure 2 (c), the proposed method is sensitive to local epoch hyperparameter because it assumes that local models converge. However, most experiments are conducted at a local epoch of 20, and such a large local epoch makes comparisons with baselines unfair. It is of course an advantage that the performance of the proposed method is good even when the local epoch is large, but this does not mean that simply increasing the local epoch is always good.
> In addition to the local epoch, I believe that other hyperparameters used in the experiment are not suitable for comparison with baselines.
> For example, the learning rate was set to 1e-3, which helps the local model to converge quickly with a large local epoch. However, it is unclear whether such a small learning rate and large local epoch are suitable for other baselines.
> Based on the code included in the supplementary material, I reproduced the FedAvg experiment of CIFAR100 - Dirichlet (0.5) in Table 1. Using the same hyperparameter presented in the paper, FedAvg showed an accuracy of about 38%, which is almost similar to 38.86% recorded in Table 1. However, when lr is 0.1, weight decay is 1e-3, local epoch is 5 and the same experiment was performed with SGD without momentum, the accuracy was about 49%, which is almost similar to the proposed method (49.31%). This does not mean that the proposed method shows similar performance to FedAvg, but I just think that the hyperparmaters should have been chosen a little more carefully. (If possible, can you run the experiment again with the hyperparameters mentioned above?)
> ```
>
> **R2**. Thanks for raising the concern.
>
> *Our algorithm is not sensitive to the number of local epochs, but benefits from it.* Considering that the communication cost is generally the bottleneck of federated learning, many works seek to reduce the number of rounds by benefiting from the increasing of local epochs [1,2,3]. Figure 2(c) presents that our algorithm has this advantage. To reach the same accuracy, our algorithm requires fewer communication rounds with larger local epochs. As a comparison, FedAvg does not necessarily achieve improvements as the number of local epochs increases. Besides, as shown in Figure 2(c) and (d), our algorithm always outperforms FedAvg over various local epochs. In fact, many existing works also adopt the 20 epochs as the basic configuration [4,5].
>
> To demonstrate the effectiveness of our proposed method over a broader range of hyper-parameters. We conduct more experiments by using a different learning rate and smaller local epochs. The results are shown in the following three tables.
>
>
> Table 1. The learning rate is 0.1, weight decay is 1e-3, local epoch is 5, and the optimizer is SGD without momentum.
>
> | Method| Accuracy|
> |--|--|
> |FedAvg|48.20% (±0.42%)|
> |FedProx|54.46% (±1.01%)|
> |FedExP|57.00% (±1.02%)|
> |FedSAM|57.94% (±0.36%)|
> |FedDF|58.83% (±0.81%)|
> |FedGEN|55.29% (±1.37%)|
> |FedMA| 49.45% (±0.61%)|
> |GAMF| 50.87% (±0.27%)|
> |FedLAW|51.78% (±0.97%)|
> |FedCDA|**65.50%** (±0.25%)|
>
>
> Table 2. The learning rate is 0.1, weight decay is 1e-3, local epoch is 10, and the optimizer is SGD without momentum.
>
> | Method | Accuracy |
> | :--: | :--: |
> | FedAvg | 46.92% (+- 0.72%) |
> | FedCDA| **66.25%** (+- 0.25%) |
>
> Table 3. The learning rate is 0.1, weight decay is 1e-3, local epoch is 20, and SGD without momentum.
>
> |Method|Accuracy|
> | :--: | :--: |
> |FedAvg|45.01% (+- 0.81%) |
> |FedCDA|**66.62%** (+- 0.25%) |
>
> We can see that our method still achieves great improvement under hyperparameters of the learning rate being 0.1 and the number of local epochs being 5, 10, and 20. In fact, a larger local learning rate helps the convergence of local training, where this configuration is more suitable for our method.
>
> We are very pleased that you have tested our code of FedAvg. **We sincerely hope that you can further directly test our proposed algorithm under various configurations.** We believe that the results you obtain are more convincing than our claim.
>
> [1] Yan Sun, et al. Fedspeed: Larger local interval, less communication round, and higher generalization accuracy. ICLR 2023
>
> [2] Sebastian U. Stich: Local SGD Converges Fast and Communicates Little. ICLR 2019
>
> [3] 	Konstantin Mishchenko, et al. ProxSkip: Yes! Local Gradient Steps Provably Lead to Communication Acceleration! Finally! ICML 2022
>
> [4] Xin-Chun Li, et al. Federated Learning with Position-Aware Neurons. CVPR 2022
>
> [5] Hongyi Wang, et al. Federated Learning with Matched Averaging. ICLR 2020
>
> ```
> W3. The overall writing quality of the paper is not good. There are many incorrect parts in the notation related to formulas and figures in the paper, and many sentences require revision.
> ```
> **R3**. Thanks for pointing out the limitations of our presentation. We have revised the presentation throughout the paper. For example, we refine the motivation example of Figure 1. We are glad that you would like to provide some suggestions for further improvement.

---

> ### Author Response · Authors · 2023-11-21
> **Looking forward to your feedback**
>
> Dear Reviewer rxU6, we thank you once again for recognizing the novelty and theoretical analysis of our work.
>
> We have provided responses to all your questions and concerns. More specially, we have conducted a series of experiments over experiments with different hyper-parameters. All these results are added to our revised manuscript. We would like to know if these responses solve your concerns. We look forward to receiving your reply eagerly.

---

> > ### Comment · Reviewer_rxU6 · 2023-11-22
> >
> > I read the rebuttal carefully and thank the authors for addressing my concerns.
> > The experimental results in Table 2 make the paper more convincing.
> > (BTW, the corresponding paragraph in the revised paper has a typo. 49.38% => 47.38%)
> > Thus, I increased my rating to 6 (marginally above the acceptance).

---

> > > ### Author Response · Authors · 2023-11-22
> > > **Thank you very much for your response**
> > >
> > > Heartfelt thanks for your response!
> > > We have corrected the typo in the refined manuscript.

---

### Official Review · Reviewer_jkW8 · 2023-11-01

**Soundness:** 2 fair
**Presentation:** 2 fair
**Contribution:** 2 fair
**Rating:** 5
**Confidence:** 4

**Summary:**

This paper focus on designing a model aggregation strategy for federated learning. Different from the traditional methods that aggregate local models from the current round only and may cause diversity issues, this paper introduces FedCDA, which selectively combines models from different rounds to maintain knowledge diversity. Experiments show that FedCDA outperforms the existing model aggregation methods.

**Strengths:**

1. This paper is easy to follow.
2. To the best of the reviewer's knowledge, the approach presented in this paper, which involves the selective aggregation of local models from various rounds to mitigate disparities between them, appears to be relatively novel.
3. The author conducted an extensive series of comparative experiments.

**Weaknesses:**

1. First of all, the motivation about Figure 1 is difficult to understand, where the authors attempt to illustrate the usefulness of cross-round local models. But according to the reviewer's understanding, the model aggregation strategy of Fig. 1(b) will cause the global model to overfit to the local node, resulting in extremely unstable global convergence. Further clarification is needed on why it is better to update model via Fig. 1(b).
2. Why the optimization objective function presented in Eq. (2) is derived based on the effectiveness of cross-round local models remains unclear. The connection between the optimization objective and the effectiveness of cross-round local models requires further elucidation.
3. Another issue lies in the optimization objective presented in Eq. (7) on the server side, where model derivatives appear to be computed for inactivate nodes. Question then arises as to whether these non-active models are stored on the server side. In federated scenarios characterized by a large number of nodes and typically large models, this could lead to substantial storage overhead.

**Questions:**

In Theorem 3, the authors claimed that convergence is achievable but did not provide a specific rate of convergence. It would be beneficial for the authors to elucidate the relationship between the convergence rate and parameter $T$ to enhance the assessment of the theoretical contribution when compared to other algorithms.

*Post-rebuttal comments:*

While some of my concerns have been addressed, several questions still remian.

1. The author's depiction of motivation in Figure 1 still appears somewhat speculative. The clients trained based on local data, and the rationale behind the considerable variation in the direction of updates across clients in different rounds remains unclear.

2. The proposed method incurs a substantial storage cost, significantly constraining the practicality of the algorithm.

Overall, I will maintain the original score.

---

> ### Author Response · Authors · 2023-11-16
> **Responses to weakness and questions**
>
> Weaknesses:
> --------------
> ```
> W1. First of all, the motivation about Figure 1 is difficult to understand, where the authors attempt to illustrate the usefulness of cross-round local models. But according to the reviewer's understanding, the model aggregation strategy of Fig. 1(b) will cause the global model to overfit to the local node, resulting in extremely unstable global convergence. Further clarification is needed on why it is better to update model via Fig. 1(b).
> ```
> **R1**. Thanks for your comment that indicates that our illustration may cause misunderstandings. We have refined Figure 1 and its corresponding description, which can be found in the revised manuscript.
>
> In the cross-round method of our example, the global model $w_{t+1}$ is the aggregation of the $t$-th round local model $w_{t}^1$ of client 1 and the $t+1$-th round local model $w_{t+1}^2$ of client 2. It does not overfit to any specific client. In the single-round method, the global model $w_{t+1}$ is the aggregation of the local model $w_{t+1}^1$ of client 1 and $w_{t+1}^2$ of the client 2 in the same $t+1$-th round.
>
> $w_{t+1}$ obtained through cross-round aggregation is close to the local optimum 1 of client 1 and the local optimum 2 of client 2, while the single-round aggregation is distant from any local optima. Therefore, $w_{t+1}$ obtained through cross-round aggregation can fit the data of all clients.
>
> ```
> W2. Why the optimization objective function presented in Eq. (2) is derived based on the effectiveness of cross-round local models remains unclear. The connection between the optimization objective and the effectiveness of cross-round local models requires further elucidation.
> ```
>
> **R2**. Thank you for pointing out the limited explanation in this part.  We have made revisions to present the derivation process of Eq. (2) and connections to cross-round aggregation. It is shown in the blue texts of Section 4 of our revised manuscript.
>
> Eq. (2) (corresponding to Eq. (3) in the revised manuscript) is derived by separately applying the L-smooth assumption over the local objective function $F_n(w^n)$ of each client n. Then, after a series transformation, the objective becomes Eq. (5) in the revised manuscript. As explained by the blue text, Eq. (5) is to essentially select local models that exhibit minimal divergence across multiple rounds.
>
>
> ```
> W3. Another issue lies in the optimization objective presented in Eq. (7) on the server side, where model derivatives appear to be computed for inactivate nodes. Question then arises as to whether these non-active models are stored on the server side. In federated scenarios characterized by a large number of nodes and typically large models, this could lead to substantial storage overhead.
> ```
>
> **R3**. Thank you for raising this concern. The server needs to store K local models of all clients. Although our algorithm requires more storage of the server than FedAvg, the cost is totally manageable in practical settings:
>
> *First*, the number $K$ of cached models is small. Our evaluation shows that merely setting $K=3$ can achieve significant improvement.
>
> *Second*, the models of non-active clients can be stored in the disk. Disk is cheap and can satisfy the storage requirement of models designed for clients. Taking the commonly used ResNet-18 (44MB) and MobileNet-V3-Large (21MB) in edge devices as examples, a disk of $1h$ TB can approximately store $20,000$ local models and $40,000$ local models, respectively. Considering that a mobile phone can have 1TB storage in the current era, we believe that this storage cost is well within the budget of an aggregation node which is usually provided by a cloud.
>
> In fact, our algorithm can be applied to most FL scenarios that existing methods considered:
>
> *Cloud-edge scenario*. Cloud serves as the aggregation node and edge devices serves as clients. Generally, constrained resources of the edge devices are the bottleneck of this scenario while the capability of the cloud is strong. Our algorithm requires no extra resources of edge devices as compared to basic FedAvg and thus can be applied to this scenario.
>
> *Cross-silo scenario*. Cloud serves as both the aggregation server and clients. The participated clients are usually institutes of which its number is usually small. Hence, the memory cost of our algorithm is acceptable in such a scenario.
>
>
> Questions:
> --------------
> ```
> Q1. In Theorem 3, the authors claimed that convergence is achievable but did not provide a specific rate of convergence. It would be beneficial for the authors to elucidate the relationship between the convergence rate and parameter T to enhance the assessment of the theoretical contribution when compared to other algorithms.
> ```
>
> **A1**. Our algorithm converges with a convergence rate $O(\frac{1}{\sqrt{T}})$ in terms of T on non-convex functions. We have revised the theorem in the refined manuscript to present this convergence rate.

---

> ### Author Response · Authors · 2023-11-21
> **Looking forward to your feedback**
>
> Dear Reviewer jkW8, we thank you once again for recognizing the presentation, innovation, and evaluation of our algorithm.
>
> We have provided responses to all your questions and concerns. For example, we improve the motivation example in Figure 1 by refining the figure and its illustration. We would like to know if these responses solve your concerns. We look forward to receiving your reply eagerly.

---

### Official Review · Reviewer_Pshb · 2023-11-01

**Soundness:** 3 good
**Presentation:** 2 fair
**Contribution:** 2 fair
**Rating:** 6
**Confidence:** 4

**Summary:**

This paper introduced FedCDA to solve statistical heterogeneity issues, especially the discrepancies between local models. The principle behind FedCDA is that the local model from each client may converge to different local optima over rounds. Based on this principle, the author utilizes a local model from multiple rounds to minimize the divergence from other clients to make sure that the aggregated global model remains aligned with selected local models. Empirical results show their algorithm’s superiority to other benchmark methods.

**Strengths:**

1. The proposed algorithm addressed the data heterogeneity issues from a novel view, considering the aggregation of cross-round local models.
2. This paper provides theoretical insights and the convergence bound of the proposed method.

**Weaknesses:**

1.	In FedCDA, each local client needs to store a set of recent K local models, which requires a much higher demand of memory space compared to other FL methods.
2.	Although FedCDA employed the approximation strategy, communicating the model set is still too heavy and the considerable extra communication may be a severe problem when there exists a large number of engaging clients.

**Questions:**

1.	Why not compare with those methods that personalize partial model parameters for each client? Personalized Federated Learning can help local models adapt to local data distribution.
2.	FedCDA maintains a consistent global model that will not interfere with those outlier local models as the authors selected the closer local model to do aggregation. The potential problem is that the aggregated model may not adapt to those outliers and be quite sub-optimal.

---

> ### Author Response · Authors · 2023-11-16
> **Responses to weakness and questions**
>
> Weaknesses
> -----------------
>
> ```
> W1. In FedCDA, each local client needs to store a set of recent K local models, which requires a much higher demand of memory space compared to other FL methods.
> ```
>
> **R1**: Thank you for your careful reviewing. These K local models are stored in the server instead of the client. Our algorithm does not require that the client has a high storage capability. Although our algorithm consumes more memory of the server than FedAvg, the cost is totally manageable in practical setting:
>
> *First*, the number $K$ of cached models is small. Our evaluation shows that merely setting $K=3$ can achieve significant improvement.
>
> *Second*, the models of not-sampled clients can be stored in the disk. Disk is cheap and can satisfy the storage requirement of models designed for clients. For example, a disk of $1$ TB can approximately store $20,000$ ResNet-18 models which are generally used in edge devices and costs $44$ MB. Given that a mobile-phone is even equipped with $1$ TB storage, we believe that the cost can be ignored by the aggregation node which is usually served as by a cloud.
>
> In fact, our algorithm can also be applied to most FL scenarios that existing methods considered:
>
> *Cloud-edge scenario*. Cloud serves as the aggregation server and edge devices serves as clients. Generally, constrained resources of the edge devices are the bottleneck of this scenario while the capability of the cloud is strong. Our algorithm requires no extra resources of edge devices as compared to basic FedAvg and thus can be applied to this scenario.
>
> *Cross-silo scenario*. Cloud serves as both the aggregation server and clients. The participated clients are usually institutes of which its number is usually small. Hence, the memory cost of our algorithm is acceptable in such a scenario.
>
>
> ```
> W2. Although FedCDA employed the approximation strategy, communicating the model set is still too heavy and the considerable extra communication may be a severe problem when there exists a large number of engaging clients.
> ```
>
> **R2**: Thanks for raising the concern. Our algorithm has the same communication cost as the basic FedAvg, i.e., broadcasting a global model to all sampled clients and receiving a single local model from each sampled client. These past K local models are stored in the server.
>
>
>
> Questions:
> -----------------
>
> ```
> Q1. Why not compare with those methods that personalize partial model parameters for each client? Personalized Federated Learning can help local models adapt to local data distribution.
> ```
>
> **A1**. Our objective is inconsistent with personalized federated learning. Training a global model and training personalized models are two different directions in FL. They have different applicable scenarios. For example, disease diagnosis requires training a global model while item recommendation prefers training personalized models. This paper focuses on training a global model.
>
> ```
> Q2.  FedCDA maintains a consistent global model that will not interfere with those outlier local models as the authors selected the closer local model to do aggregation. The potential problem is that the aggregated model may not adapt to those outliers and be quite sub-optimal.
> ```
>
> **A2**. Thank you for raising such an insightful comment. We agree that our algorithm cannot exclude all outliers. But, as compared to existing aggregation methods, our algorithm can greatly mitigate this issue via an aggregation manner of minimizing model divergence. Besides, our objective is to train a global model which has high accuracy over the global dataset. Although the global model cannot well adapt to these outliers, it still can have high global accuracy.

---

> ### Author Response · Authors · 2023-11-21
> **Looking forward to your feedback**
>
> Dear Reviewer Pshb, we appreciate your reviewing once again.
>
> We have provided responses to all your questions and concerns, especially including the discussion about the memory and communication costs. We would like to know if these responses solve your concerns. We look forward to receiving your reply eagerly.

---

### Official Review · Reviewer_erYc · 2023-11-07

**Soundness:** 3 good
**Presentation:** 3 good
**Contribution:** 3 good
**Rating:** 6
**Confidence:** 3

**Summary:**

Traditional federated learning aggregation methods only consider the local model from the current round, failing to account for differences between clients. The proposed FedCDA method selectively aggregates local models from multiple rounds for each client to decrease discrepancies between clients. By aggregating models that remain aligned with all clients' data, FedCDA can better preserve each client's local knowledge in the global model, outperforming state-of-the-art aggregation baselines as shown in experiments.

**Strengths:**

The paper is well organized, and it is mostly easy to follow the different parts.
This paper makes the multi-epoch aggregation model problem equivalent to a combinatorial optimization problem.
The method seems novel to me, however there may be other methods that include a similar mechanism which I have missed.
Sufficient theory illustrations well explain the objective of the proposed FedCDA.

**Weaknesses:**

- The variables in the formula are very complex, and a lot of space was spent on deriving the objective function. At the same time, it is still difficult to understand why selectively aggregating them can make the global model approach local models more closely and affect the targeting of local data knowledge?
- It seem that the caption of Fig(2) is wrong.
- There seems to be a lack of discussion on memory and communication costs.

**Questions:**

- How to understand that there are varied received global models and they result in local models? (Quote from Introduction: "each client often converges to different models due to varied received global models and multiple local optima of deep neural networks”?)
- Do the local models achieve different local minimal only due to their own private local data?
- Why are there varied received global models?
- Authors claim that selectively aggregating them can make the global model approach local models more closely and affect the forgetting of local data knowledge.
- Does Equation (2) have used the approximation (∇wn Fn(wn))’ ≈ 0 ?
- What is the meaning of N-P+bP/B? why add the #nums unselected N-P?
- why selectively aggregating them can make the global model approach local models more closely and affect the targeting of local data knowledge?
- Authors seem to use the random sampling strategy to minimize the eq.7, i.e. selectively aggregating multi-epoch local models, what if P=N?
- The authors aim to make the global model approach any of the different local models in multiple rounds, how do define the “approach”, and how approach between the multi-local epoch global model and last-epoch local models?
- What differences between this work “Understanding How Consistency Works in Federated Learning via Stage-wise Relaxed Initialization. Dacheng Tao et.al .Nips2023.”

---

> ### Author Response · Authors · 2023-11-16
> **Responses to weakness**
>
> ```
> W1. The variables in the formula are very complex, and a lot of space was spent on deriving the objective function. At the same time, it is still difficult to understand why selectively aggregating them can make the global model approach local models more closely and affect the targeting of local data knowledge?
> ```
>
> **R1**: Thanks for your comment that indicates that the presentation of our motivation is not clear enough. We have refined Figure 1 and its corresponding description, which can be found in the revised manuscript.
>
> As shown in the Figure 1.(b), in the cross-round method, the global model $w_{t+1}$ is the aggregation of the $t$-th round local model $w_{t}^1$ of client 1 and the $t+1$-th round local model $w_{t+1}^2$ of client 2. In the single-round method, the global model $w_{t+1}$ is the aggregation of the local model $w_{t+1}^1$ of client 1 and $w_{t+1}^2$ of the client 2 in the same $t+1$-th round.
>
> $w_{t+1}$ obtained through cross-round aggregation is close to the local optimum 1 of client 1 and the local optimum 2 of client 2, while the single-round aggregation is distant from any local optima. Therefore, by referring to the concepts of the field of continual learning where the knowledge of old model will be maintained when the new model is close to the old model, we claim that the global model $w_{t+1}$ obtained through cross-round aggregation can maintain the local data knowledge of local models.
>
> ```
> W2. It seem that the caption of Fig(2) is wrong.
> ```
>
> **R2**. Thank you very much for the careful reviewing. We have reordered the captions to match their corresponding sub-figures.
>
> ```
> W3. There seems to be a lack of discussion on memory and communication costs.
> ```
>
> **R3**. Thanks for raising the concern. The memory costs have been discussed by the Proposition 1, 2, 3 and the last-second paragraph before section 4.1 of our original paper. As compared to FedAvg which only caches $1$ model of each client sampled in current round, our algorithm requires the server to cache $K$ models for all clients respectively. Although the memory cost is several times than FedAvg, our algorithm is also totally manageable in a practical setting.
>
> *First*, the number of cached models $K$ is small. Our evaluation shows that merely setting $K=3$ can achieve significant improvement.
>
> *Second*, the models of not-sampled clients can be stored in the disk. Disk is very cheap now and can satisfy the storage requirement of models designed for clients. For example, a disk of $1$ TB can approximately store $20,000$ ResNet-18 models which are generally used in edge devices and costs $44$ MB. The same disk can store more models when the model is MobileNet designed specifically for devices. Given that a mobile phone is even equipped with $1$ TB storage, we believe that the cost is within the budget of the aggregation node which is usually served by a cloud.
>
> It is worthwhile to note that our algorithm has the same communication cost as the basic FedAvg, i.e., broadcasting a global model to all sampled clients and receiving a single local model from each sampled client. Our algorithm does not modify the sampling strategy and the interaction strategy of FedAvg.

---

> ### Author Response · Authors · 2023-11-16
> **Answers to questions Q1-Q6**
>
> ```
> Q1. How to understand that there are varied received global models and they result in local models? (Quote from Introduction: "each client often converges to different models due to varied received global models and multiple local optima of deep neural networks”?)
> ```
>
> **A1**. We view the local training stage of each round in a client as a complete training process corresponding to the centralized setting. The initialized model of each local training stage is the global model. Since the parameters of the global model are different across rounds, the initialized models are different in different local training stages. Considering the model may converge to local optima close to its initialized value when there are multiple local optima of the DNN, the converged local model naturally varies across rounds.
>
> ```
> Q2. Do the local models achieve different local minimal only due to their own private local data?
> ```
>
> **A2**. We consider that there are three main factors that affect the convergence of local models of different clients.
>
> First, heterogeneous data. Dataset formalizes the loss function. It is necessary to train different parameters to fit different data distributions.
>
> Second, different parameters of the global model across rounds. Different clients are sampled in different rounds and thus they perform the local training process with different global models.
>
> Third, stochastic training process. The local training process usually adopts SGD or its variants, which also affects the convergence of local minima.
>
> ```
> Q3. Why are there varied received global models?
> ```
>
> **A3**: The “varied global models” means that the parameters of the global model of different rounds are different. We have revised the corresponding texts in the abstract and introduction in the manuscript, which are highlighted by the blue color.
>
> For example, to avoid ambiguity, the corresponding text in the abstract is revised to be “The principle behind FedCDA is that due to the different global model parameters received in different rounds and the non-convexity of deep neural networks, the local models from each client may converge to different local optima across rounds”.
>
> ```
> Q4. Authors claim that selectively aggregating them can make the global model approach local models more closely and affect the forgetting of local data knowledge.
> ```
>
> **A4**. This concept of maintaining knowledge is borrowed from the field of continual learning, where the knowledge of old model will be maintained when the new model is close to the old model. The global model obtained through our cross-round aggregation method is close to local models and thus, can maintain their local data knowledge.
>
> ```
> Q5. Does Equation (2) have used the approximation (∇wn Fn(wn))’ ≈ 0 ?
> ```
>
> **A5**: No.  Equation (2) is derived based on the L-smooth assumption. The detailed derivation process can be found in the revised manuscript, which is denoted by the blue color.
>
> ```
> Q6. What is the meaning of N-P+bP/B? why add the #nums unselected N-P?
> ```
>
> **A6**. Our objective is to find a global model that is close to local models of all clients. As a consequence, we incorporate the local models of these unselected clients into the selection process. However, considering the potentially high computation cost of selecting local models from all clients, for the unselected clients, we propose only utilizing their local models selected in the previous round, which have already been through the selection process and are close to each other.

---

> ### Author Response · Authors · 2023-11-16
> **Answers to Q7-Q10**
>
> ```
> Q7. why selectively aggregating them can make the global model approach local models more closely and affect the targeting of local data knowledge?
> ```
>
> **A7**. Considering the aggregation of the two-client FL, given the local model $w^1_t$ of the client 1 and two local models ${w^2_{t-1}, w^2_{t}}$ of the client 2, there are two aggregation manners, i.e., $w_t = 1/2(w^1_t+ w^2_{t-1})$ and $w_t = 1/2(w^1_t+ w^2_{t})$. Choosing which combination manner relies on their distance. Specifically, if $w^1_t$ is closer to $w^2_{t-1}$ than $w^2_{t}$, then the first aggregation manner is adopted. Otherwise, the second manner is adopted. In this way, the obtained global model $w_t$ is close to at least one local model of each client. When merely adopting existing aggregation methods, i.e., the second aggregation manner, the obtained global model $w_t$ may be far from any local models of all clients. More details can also be found in the response R1 for the weakness W1.
>
> ```
> Q8. Authors seem to use the random sampling strategy to minimize the eq.7, i.e. selectively aggregating multi-epoch local models, what if P=N?
> ```
>
> **A8**. When P=N, our algorithm will randomly group N clients into B batches and then, it performs selective aggregation batch by batch with (7).
>
> ```
> Q9. The authors aim to make the global model approach any of the different local models in multiple rounds, how do define the “approach”, and how approach between the multi-local epoch global model and last-epoch local models?
> ```
>
> **A9**. The “approach” is measured as the variance among local models, i.e., the average of L2-norm distance of the global model to all local models. This description can be found in the eq.(5) and (6) of the revised manuscript.
>
> In the question of “how approach between the multi-local epoch global model and last-epoch local models”, we guess you may be asking about “how approach between the global model obtained by cross-rounds aggregation and the local models of the latest round”. Unfortunately, the model distance can hardly be analyzed due to the unknown distribution of local optima. Yet, our paper has shown that the difference between their losses is bounded, as shown in Lemma 1 of Appendix B.4.
>
> ```
> Q10. What differences between this work “Understanding How Consistency Works in Federated Learning via Stage-wise Relaxed Initialization. Dacheng Tao et.al .Nips2023.”
> ```
>
> **A10**. FL consists of two main steps: step 1 of local update and step 2 of global aggregation. This related work focuses on step 1 by applying an offset to the local model, while our work focuses on step 2 by applying a cross-round aggregation of all local models. Our method is orthogonal to the related work and can be jointly applied to FL together with this work. We have discussed the difference in the first paragraph of the related work section in our revised manuscript.

---

> > ### Author Response · Authors · 2023-11-21
> > **Looking forward to your feedback**
> >
> > Dear Reviewer erYc, we appreciate your reviewing once again.
> >
> > We have responded to all your questions and concerns. We also improved the presentation of the motivation in the main text. We would like to know if these responses solve your concerns. We look forward to receiving your reply eagerly.

---

### Author Response · Authors · 2023-11-16
**Responses to common concerns**

We would like to express our sincere gratitude to all reviewers for the valuable comments which have significantly improved the quality of our paper. We are delighted that the novelty of our approach has been recognized by all reviewers and that our theoretical contributions have also received recognition from Reviewer erYc, Pshb, and rxU6. Nevertheless, there are also some common concerns. In addition to providing responses for each reviewer, we have also summarized them here for ease of reading.

```
W1. The motivation of why selectively aggregating local models can make the global model approach local models more closely is unclear.
```

**R1**. We here present a simplified example for ease of understanding. More details can also be found in the illustration of Figure 1 in our manuscript.

Considering the aggregation of the two-client FL, given the local model $w^1_t$ of the client 1 and two local models ${w^2_{t-1}, w^2_{t}}$ of the client 2, there are two aggregation manners: first, $w_t = 1/2(w^1_t+ w^2_{t-1})$ and second, $w_t = 1/2(w^1_t+ w^2_{t})$.

Choosing which combination manner of our algorithm relies on the model distance. Specifically, if $w^1_t$ is closer to $w^2_{t-1}$ than $w^2_{t}$, then the first aggregation manner is adopted. Otherwise, the second manner is adopted. In this way, the obtained global model $w_t$ is close to at least one local model of each client. When merely adopting existing aggregation methods, i.e., the second aggregation manner, the obtained global model $w_t$ may be far from any local models of all clients.





```
W2. Since the server has to cache the local models for clients, the memory cost may be large.
```

**R2**. The memory cost of our algorithm is totally manageable in FL practical settings.

*First*, the number $K$ of cached models is small. Our evaluation shows that merely setting $K=3$ can achieve significant improvement.

*Second*, the ratio of participated clients is usually small while the local models of non-participated clients can be stored on the disk which has sufficient storage space. For example, a $1$TB hard drive can store approximately $20,000$ copies of ResNet-18, which is widely adopted on the edge. As considering MobileNet which is commonly used in edge devices, the server can store more copies (approximately $40,000$). Given that even a mobile phone is equipped with $1$ TB storage, we believe that the cost is within the budget of the aggregation node which is typically hosted by a cloud.

In reality, our algorithm can also be applied to most FL scenarios that existing methods take into consideration.

*Cloud-edge FL scenario*. Cloud serves as the aggregation server and edge devices serves as clients. Generally, constrained resources of the edge devices are the bottleneck of this scenario and the models adopted in these edge devices are usually small. Considering the capability of the cloud is strong, the memory cost of our algorithm is within the budget.

*Cross-silo FL scenario*. Cloud serves as both the aggregation server and clients. The participants are usually institutes with a small number. Hence, the memory cost of our algorithm is acceptable in such a scenario.



```
W3. The communication cost is large.
```

**R3**. Our algorithm has the same communication cost as the basic FedAvg, i.e., broadcasting a global model to all participated clients and receiving a single local model from each participated client. Our algorithm does not bring extra communication cost.

---

### Author Response · Authors · 2023-11-22
**A summary of revised parts of our manuscript**

Dear reviewers and chairs,

We express our gratitude once again for your thorough review of our papers. To enhance readability, we here provide a summary of the revised sections in the manuscript. The major concerns center around the motivation example and the memory cost of our algorithm. Consequently, our primary revisions are outlined below.


Firstly, we enhance the clarity of Figure 1 in the motivational example, along with its accompanying illustration. Additionally, we introduce Formula (2) to facilitate a better understanding of the origin of the cross-round optimization objective.

Secondly, we augment the discussions regarding memory cost in the concluding paragraph of Section 4.

In addition to the aforementioned revisions, we have further polished the writing across the entire paper and incorporated evaluations conducted with an expanded set of hyperparameters.


Thanks,

Authors

---

### Meta-Review · Area_Chair_aYgp · 2023-12-22

**Metareview:**

This paper considers the aggregation component of federated learning, specifically proposing to selectively aggregate over local models from multiple rounds per client. This is done by considering the objective of reduced discrepancies between the clients. A range of theoretical and empirical results are provided, including complexity and accuracy improvements.

  The reviewers appreciated the interesting point of view for addressing heterogeneity, and the fact that the empirical results are paired with theoretical insights. However, a number of concerns were raised including the high memory demand of storing K local models, the intuition and writing of the paper and lack of polish, lack of experiments with more clients and different participation ratios, and most notably the lack of consistent comparison across hyper-parameters. The authors presented a rebuttal containing a number of clarifications (including convergence rate, etc.), additional experiments with different participation rates and hyper-parameters, and argument for the practicality of assuming such a large amount of memory requirements.

  Overall, some reviewers raised their scores although one reviewer still rated slightly below acceptance. However, after reviewing all of the materials (paper, reviews, rebuttal, and discussion) it seems that the rebuttal does address all of the reviewers' concerns. As a result, I recommend acceptance.

**Justification For Why Not Higher Score:**

Based on the reviews, scores, and impact, I don't believe this paper warrants a higher score.

**Justification For Why Not Lower Score:**

This paper could be lowered to a reject, given the number of concerns; however, it does seem to me that the rebuttals address all of the points made by the reviewers, except perhaps for the reasonableness of the large memory requirements which is arguable.

---

### Decision · Program_Chairs · 2024-01-16

Accept (poster)